# Traits and Transports: The Effects of Personality on the Choice of Urban Transport Modes

**John Magnus Roos** [1,2,3,4,*], **Frances Sprei** [2]  **and Ulrika Holmberg** [1]

1   Centre for Consumption Research, School of Business, Economic and Law, University of Gothenburg,
    SE-405 30 Gothenburg, Sweden; ulrika.holmberg@cfk.gu.se
2   Division of Physical Resource Theory, Chalmers University of Technology, SE-412 96 Gothenburg, Sweden;
    frances.sprei@chalmers.se
3   School of Health Sciences, University of Skövde, SE-541 28 Skövde, Sweden
4   Department of Business Administration and Textile Management, University of Borås,
    SE-501 90 Borås, Sweden
*   Correspondence: magnus.roos@his.se or magnus.roos@cfk.gu.se

**Featured Application: Traffic noise and air pollution caused by car traffic in urban areas can be reduced by improving bicycle and public transportation options. To develop sustainable cities, it is important to focus on people, for instance how they move around and socialize during traveling. The present study highlights some differences between people who use different travel modes. Most interesting–the personality is almost the opposite between people who frequently use public transportation and people who frequently use cars. Sustainable urban planners and policymakers can apply these insights to communicate and design for the well-being of their citizens.**

**Abstract:** We examine the influence of personality on car driving, usage of public transport and cycling. Personality is measured through the Big Five personality traits (i.e., Openness, Conscientiousness, Extraversion, Agreeableness and Neuroticism) and Environmental personality. Data were collected through a Web-based panel of adult citizen in the city of Gothenburg, Sweden ($N = 1068$). Age, gender, income, children at home and residential area were used as control variables. Car driving is influenced by low degree of Openness, high degree of Conscientiousness, and low degree of Environmental personality. Usage of public transport is influenced by low degree of Conscientiousness, high degree of Agreeableness, and high degree of Environmental personality. Cycling is foremost influenced by a high degree of Environmental personality.

**Keywords:** urban travel habit; sustainable urban planning; transportation mode; personality





## 1. Introduction

Sweden has set the goal to reduce carbon emissions in the transport sector by 70% from 2010 to 2030 and reach net neutrality by 2045. One strategy to reach this goal might be to change transport behavior from individual car use to more sustainable transport modes, such as public transportation and cycling. In order to shift toward more sustainable mobility patterns, we need to better understand the users of the different modes.

From previous studies we know that the variables that have the strongest effect on people's transport choices are travel time and cost [1]. In addition, demographic variables, especially gender and age, but also income, presence of children and residential area, influence preferences and usage of different transportation modes, for instance car, bike, and public transport [2–4].

The present study will go beyond such well-studied variables and instead focus on travel behavior in relation to personality traits. Personality traits are included as

a multitrait approach (i.e., the Big Five personality traits), and a single-trait approach (Environmental personality).

Compared to the other variables mentioned above, personality traits have been less studied in relation to usage of transportation modes, especially public transport and cycling. Regarding car use, personality traits have found to be associated with driving style and risky behavior [5], as well as choices of different car brands [6]. Almost all these studies conclude that assessing personality traits (such as Openness, Conscientiousness, Extraversion, Agreeableness and Neuroticism) are important to understand car usage. Therefore, it is surprising that so few studies have relied on personality traits in modeling travel behavior in relation to different modes. Especially since travel behavior is relatively stable and less influenced by situational factors [7–11].

Personality traits may be promising to consider as a determinant of travel behavior and choice of different modes. Car usage is interesting to study since it is the most common mode in Sweden and other Western countries. In order to change transport behavior from car driving toward more sustainable options, it might be of interest to study public transportation and cycling in relation to the same personality traits. The purpose of the present study is to examine to what extent travel behavior can be explained by personality traits–both the Big five personality framework and Environmental personality. For this we formulated three research questions:

Research Question 1: How is frequency of car driving affected by the Big five personality traits and Environmental personality?

Research Question 2: How is frequency of use of public transportation affected by the Big five personality traits and Environmental personality?

Research Question 3: How is frequency of cycling affected by the Big five personality traits and Environmental personality?

## 2. Definition of Personality Traits

Personality is a pattern of relatively permanent factors that give both consistency and individuality to a person's behavior [12–14].

### 2.1. The Big Five Personality Traits

The Big Five model is a widely accepted personality model [15,16]. It posits that there are five major and universal traits of personality: Openness, Conscientiousness, Extraversion, Agreeableness and Neuroticism.

Openness (sometimes referred to as Openness to experiences) is associated with behavioral flexibility, intellectual curiosity, aesthetic sensitivity, vivid imagination, and unconventional attitudes [15]. People high on Openness are open for innovations, different cultures and emotions of other people. They have broad interests and are imaginative. People low on Openness are cautious and conservative [12,13].

Conscientiousness is associated with responsibility, self-discipline, order, competence, and dutifulness [15]. People high on Conscientiousness are thorough and efficient. They are forward thinking, self-controlled and plan their lives. The order facet of Conscientiousness is positively related to traditionalism [17]. People low on Conscientiousness are easy-going, impulsive, and lazy. They usually act on the spur of the moment [12,13].

Extraversion is associated with social behavior, high activity, experience of positive affect, well-being, impulsiveness, and assertiveness [15]. People high on Extraversion are sociable and outgoing. They have a greater impact on their social environment. People low on Extraversion are reserved and withdrawn. They tend to be more similar to wallflowers [12,13].

Agreeableness is associated with prosocial behavior, friendliness, trust, altruism, and tender mindedness [15]. People high on Agreeableness are compassionated, trusting, and forgiving. They are good negotiators to solve conflicts and strive for agreements in which all cooperate. People low on Agreeableness are suspicious and argumentative. They are

antagonistic and aggressive and seem to put themselves into a lot of social conflicts. They assert their authority and power to solve social conflicts [12,13].

Neuroticism is associated with anxiety, vulnerability, tension, worrying, and low self-confidence [15]. People high on Neuroticism are anxious and nervous. They experience a lot of mood-swings and feel anxious, tensed, and stressed. People low on Neuroticism are emotionally stable. They are composed, relaxed, and calm [12,13].

### 2.2. Environmental Personality

Environmental psychologists have attempted to characterize the "pro-environmental individual", sometimes referred to as the "Environmental personality trait" [18]. This is the person who demonstrates a pattern of pro-environmental action across time, space, and different domains, such as energy use, water consumption, transportation, waste reduction, composting and recycling habits [17,18]. Many of these behaviors have been discussed in relation to an empathic or altruistic personality [17]. Previously reported findings have suggested that Environmental personality is positively associated with several Big Five personality traits (i.e., Openness, Agreeableness, Neuroticism, and Conscientiousness) [19–21].

### 3. Previous Research

### 3.1. Personality and Choice of Modes of Transport

Regarding the Environmental personality, previous research has found that a high degree of Environmental personality increases the likelihood of choosing an environmentally friendly mode, for instance bicycling instead of public transport and public transport instead of car [22].

Previous research regarding the Big Five personality traits and choice of mode of transport is very limited. To the best of our knowledge there are only two studies that investigate the Big Five personality traits in relation to car driving and use of public transportation. The present study seems to be the first study to report results from the Big Five personality traits in relation to bicycling.

A nationally representative study of the Swedish population reports that car driving is influenced by a low degree of Openness, a high degree of Extraversion and a low degree of Neuroticism [23]. The same study reported that use of public transportation is influenced by a high degree of Openness and Agreeableness, and a low degree of Conscientiousness and Extraversion [23]. In contrast, a study on the Big Five personality traits and use of public transportation in Tehran (Iran) reports that use of public transportation is positively related to Extraversion and negatively related to Neuroticism [1]. The different results can be explained by different samples. The first study represents the Swedish population based on choice of modes of transport in everyday life [23]. The respondents in the second study are departure passengers waiting at the check-in lounge, Imam Khomeini International Airport in Tehran, Iran. The second study focuses on public transportation to the airport [1].

The different results can also be explained by cultural differences. Different societies have carried out different investments regarding modes of transport and transport options also reflect the society's culture, economy, politics, technology, etc. [24,25].

### 3.2. Other Variables Influencing Personality and Choice of Modes of Transport

From previous research, we have identified age, gender, income, children at home and residential area as particularly important for the choice of mode of transport. Age, gender and income have also been shown to be related to personality. Below we will briefly describe the relationship between these variables and personality as well as these variables and choice of transport mode.

Openness is positively related to income [26] and negatively related to age [27]. Conscientiousness is positively related to age [27]. Extraversion is positively related to income [26] and negatively related to age [27]. Agreeableness is positively related to age [19] and being female [28] and negatively related to income [26]. Neuroticism is positively related to being female [28] and negatively related to age [27] and income [26].

Environmental personality is positively related to being female and income and negatively related to age [17,29,30].

Car driving is positively related to age, being male, income, having children at home and rural residential areas with low population density [18,22,23,31]. Use of public transportation is positively related to being female and urban residential areas with high population density, while negatively related to age and income [1,3,22,23,31,32]. Cycling is positively related to being male [33] and negatively related to age [25].

## 4. Materials and Methods

### 4.1. Research Design

A quantitative cross-sectional survey design is used for answering the three research questions. In reference to earlier research, behavior is theoretically driven by personality traits and not the other way around [12]. Thereby, we imply causality in analyzing the relationships between the studied personality traits and transport behavior. This is also reflected in the choice of the statistical analysis method multiple hierarchical regression.

The survey was administered online and sent to residents in the Gothenburg Area. It was designed to capture the Big Five and environmental personality traits. In the following subsections the procedure, sampling, variables, and measurements used, as well as the statistics are described more in detail.

### 4.2. Procedure

The online panel survey, the Citizen Panel at the University of Gothenburg, was used for data collection. The panel is run by the Laboratory of Opinion Research (LORE). LORE offers an infrastructure for multidisciplinary research and provides an efficient facility for collecting data from web surveys. In November 2018, there were over 58,000 registered members. The panel is not representative of the Swedish population, it is over-represented by men, older people, and people with higher income [34]. Each year, members of the panel are invited to do 2 to 3 surveys (for more information, see www.lore.gu.se, accessed on 22 January 2021).

For this study of travel behavior, 1700 members from Gothenburg and surrounding municipalities were selected to be invited as respondents.

The web-survey was collected between 12 September and 7 October 2018. In total, two reminders were sent, 12 and 18 days, respectively, after the survey was first sent to the respondents. The average time respondents spent answering the survey was 6.9 min (SD 4.4) [34].

### 4.3. Sampling

A two-stage sampling procedure was used for the data collection. In the first stage, 9 of 13 municipalities in the Gothenburg area were selected (i.e., Ale, Gothenburg, Härryda, Kungsbacka, Kungälv, Lerum, Mölndal, Öckerö, Partille). Gothenburg is by far the largest municipality and most participants were recruited from here. The other eight municipalities are surrounding municipalities with many residents commuting to Gothenburg.

In the second stage, an invitation to 1700 panel members between 18 and 74 years of age were distributed. About 4% of the gross sample turned out to have invalid or undeliverable e-mail addresses; hence, there were 1631 individuals who received the survey invitation by e-mail. In total, 1068 respondents answered the survey, giving a participation rate of 65 %.

### 4.4. Variables and Measurments

The Big Five personality traits were assessed using the Big Five Inventory BFI-10 [35], which is a 10-item inventory with 2 items measuring each personality trait. The BFI-10 has shown satisfactory levels of convergent and discriminant validity and test-retest reliability [35]. The advantage of the shorter BFI-10 is a higher response rate. Responses to the items were obtained on a four-point Likert scale ranging from 1 ("strongly disagree")

to 4 ("strongly agree"). The Openness index was constructed by averaging the responses to "has few artistic interests" (reversed) and "has an active imagination". The Conscientiousness index was constructed by averaging the responses to "tend to be lazy" (reversed) and "does a thorough job". The Extraversion index was constructed by averaging the responses to "is reserved" (reversed) and "is outgoing, sociable". The Agreeableness index was constructed by averaging the responses to "is considerate and kind to almost everyone" and "is generally trusting". The Neuroticism index was constructed by averaging the responses to "is relaxed, handles stress well" (reversed) and "gets nervous easily". Only respondents who had provided answers on both items to a specific factor were included in the subsequent analyses.

Environmental personality was measured through five items; "I carefully recycle my household waste", "In the supermarket, I choose environmentally friendly products", "I would rather buy second hand than new things", "I choose transportation modes that have as little environmental impact as possible", "It is important for me to try to repair things rather than to buy new". Each item was measured using a four-point Likert scale ranging from 1 ("strongly disagree") to 4 ("strongly agree"). When selecting the Environmental personality items, we were inspired by the Students Behavioral Environmental Scale [18]. Only respondents who had provided answers to all five items were included in the analyses.

The questions for modes of transport were "How often during the last 12 months have you?"–"Driven a car", "Traveled by public transport", "Cycled". The respondents were asked to indicate their level of usage for each one of the three transportation modes on a six-point frequency scale, ranging from 1 ("never) to 6 ("daily"). In Table 1 the frequencies of usage of the different modes are presented.

**Table 1.** Aggregated measures of the study sample (*N* = 1068).

| Control Variables | | |
|---|---|---|
| Gender | Men | 61.0% |
| | Women | 39.0% |
| Age | 18–39 years | 24.4% |
| | 40–59 years | 44.3% |
| | 60–74 years | 31.3% |
| Income | Low income | 32.2% |
| | Average income | 22.0% |
| | High income | 45.8% |
| Children in the household | | 27.7% |
| Residential area | Gothenburg municipality | 70.1% |
| | Other municipalities | 29.9% |
| Personality traits [1] | | |
| Big Five personality traits | Openness | 2.7 (0.7) |
| | Conscientiousness | 3.1 (0.6) |
| | Extraversion | 2.9 (0.7) |
| | Agreeableness | 3.1 (0.5) |
| | Neuroticism | 2.1 (0.7) |
| Environmental personality | | 2.7 (0.6) |
| Transportation mode (Usage frequency during the last 12 months) | | |
| Car as a driver | Never | 13.2% |
| | A few times the past 12 months | 10.2% |
| | A few times every month | 7.2% |
| | A few times every week | 15.4% |
| | Several times every week | 25.9% |
| | Daily | 28.1% |
| Public transportation | Never | 6.3% |
| | A few times the past 12 months | 13.2% |
| | A few times every month | 20.1% |
| | A few times every week | 15.7% |
| | Several times every week | 26.4% |
| | Daily | 18.3% |
| Cycle | Never | 27.0% |
| | A few times the past 12 months | 20.8% |
| | A few times every month | 14.5% |
| | A few times every week | 13.4% |
| | Several times every week | 15.0% |
| | Daily | 9.3% |

[1] The personality scales ranges from 1 (lowest degree) to 4 (highest degree). The values report the mean value and the standard deviation in parentheses.

Age, gender, income, children at home and residential area were used as control variables. The reasons for this were that pervious research has shown that these variables are related to personality [17,23,26–30] and choice of transport mode [1,3,22,23,25,31–33].

Gender was dummy coded; men was coded as 0 and women as 1. Age was categorized in six groups (i.e., 18–29, 30–39, 40–49, 50–59, 60–69, 70+). Income was coded into three categories; low income < 30,000 SEK; average income 30–37,000 SEK; high income > 37,000 SEK. The average monthly income in Sweden is 33,700 SEK (approximately 3450 USD or 3050 EUR) (Statistics Sweden, 2017; SEB, 2017). The presence of children in the household was dummy coded, equal to one if there are at least one child in the household, otherwise 0. Table 1 presents sample values for the demographic variables.

*4.5. Statistical Analyses*

In the beginning of the presentation of results, characteristics of the sample is presented, and the relationships between the quantitative variables are presented in a correlation matrix.

Three-step multiple hierarchical regression analyses are used for each transportation mode, respectively, to answer the research questions, in which the transport mode is the dependent variables. The first step includes the control variables (i.e., age, gender, income, children at home, residential area) as independent variables. In the second step, the Big Five personality traits are added, and in the third step the Environmental personality trait is added. The Big Five personality traits are included before the Environmental personality trait because it is a more general framework.

## 5. Results

*5.1. Descriptive Statistics*

Table 1 presents the Big Five personality factors among the participants and the mean of the Environmental personality measures in the sample. The Cronbach alpha coefficient was 0.70 ($N = 1039$) for the Environmental personality index, which is perceived as acceptable [36]. Sample characteristics regarding control variables and frequencies in car driving, use of public transport, and cycling are also reported in Table 1. Compared to the population of Gothenburg, the sample is underrepresented by females, citizens 18–39 years, people with low income, and household with children at home [37–39].

Table 2 shows the Pearson correlations between the variables. Car driving is negatively correlated with usage of public transport, while cycling is unrelated to both car driving and usage of public transport. Furthermore, as can be seen, car driving is negatively correlated with Openness, Agreeableness, Neuroticism, and Environmental personality, while positively correlated with Conscientiousness, Extraversion, and age. Usage of public transport is negatively correlated with Conscientiousness and age, while positively correlated with Openness, Agreeableness, Neuroticism and Environmental personality. Finally, cycling is negatively correlated with age, while positively correlated with Conscientiousness, Agreeableness and Environmental personality.

**Table 2.** Pearson correlation matrix for transportation modes, personality traits, and age.

| | 2 | 3 | 4 | 5 | 6 | 7 | 8 | 9 | 10 |
|---|---|---|---|---|---|---|---|---|---|
| 1. Car driving | −0.51 ** [1] | −0.02 | −0.16 ** | 0.11 ** | 0.09 ** | −0.02 | −0.17 ** | −0.19 ** | 0.22 ** |
| 2. Public transportation | | −0.03 | 0.11 ** | −0.08 * | 0.00 | 0.12 ** | 0.10 ** | 0.17 ** | −0.19 ** |
| 3. Cycling | | | 00.03 | 0.10 ** | 0.03 | 0.11 ** | 0.01 | 0.27 ** | −0.16 ** |
| 4. Openness | | | | 0.02 | 0.11 ** | 0.11 ** | 0.02 | 0.22 ** | −0.03 |
| 5. Conscientiousness | | | | | 0.21 ** | 0.09 ** | −0.18 ** | 0.13 ** | 0.03 |
| 6. Extraversion | | | | | | 0.18 ** | −0.28 ** | 0.01 | 0.15 ** |
| 7. Agreeableness | | | | | | | −0.14 ** | 0.17 ** | −0.06 |
| 8. Neuroticism | | | | | | | | 0.13 ** | −0.24 ** |
| 9. Environmental personality | | | | | | | | | −0.15 ** |
| 10. Age [2] | | | | | | | | | |

[1] * $p < 0.05$; ** $p < 0.01$. [2] Age is measured on a six-point scale (e.g., 18–29: 30–39: 40–49: 50–59: 60–69; 70-).

### 5.2. The Effects of Personality Traits on Car Driving

To examine the influence of personality traits on car driving, we conducted a hierarchical multiple regression analysis (see Table 3). Control variables were entered in step 1. the results show that frequency of car driving increases if the respondents are men, older, have an above average income and have children at home. After controlling for age, gender, income and children at home, the Big Five personality traits explained a significant amount of unique variance in car driving frequency. Low Openness and Conscientiousness were positively related to car driving. In step 3, we entered Environmental personality. People who drive cars more frequently have a lower degree of Environmental personality compared to people who drive cars less frequently.

**Table 3.** Hierarchical regression models exploring how personality predict car driving ($N = 964$).

| Independent Variables | Step 1 Demography | Step 2 Demography, Big 5 [1] | Step 3 Demography, Big 5, EP [2] |
|---|---|---|---|
| Demography | | | |
| Gender (female) | −0.18 *** [3] | −0.18 *** | −0.15 *** |
| Age | 0.23 *** | 0.22 *** | 0.21 *** |
| Income (Low) | −0.16 *** | −0.16 *** | −0.16 *** |
| Income (High) | 0.09 ** | 0.08 * | 0.08 * |
| Child in household | 0.13 *** | 0.13 *** | 0.14 *** |
| Residential Gothenburg | −0.27 *** | −0.26 *** | −0.26 *** |
| Personality | | | |
| Openness | | −0.10 *** | −0.07 ** |
| Conscientiousness | | 0.05 | 0.06 * |
| Extraversion | | 0.04 | 0.03 |
| Agreeableness | | −0.02 | 0.00 |
| Neuroticism | | 0.00 | 0.01 |
| EP | | | −0.13 *** |
| $R^2_{Adj}$ | 0.265 *** (0.269) | 0.273 *** (0.282) | 0.286 *** (0.295) |

[1] Big Five Personality traits, [2] Environmental personality. [3] Beta weights (i.e., standard regression coefficients), * $p < 0.05$; ** $p < 0.01$; *** $p < 0.001$.

### 5.3. The Effects of Personality Traits on Usage of Public Transportation

To examine the influence of personality traits on usage of public transport, we conducted another hierarchical multiple regression analysis (see Table 4). Step 1 shows that frequency of usage increases if the respondents are women, young, and do not have children at home. After controlling for these variables, the Big Five personality traits were entered in step 2 and explained a significant amount of unique variance in usage of public transport. High degree of Agreeableness and Openness and low degree of Conscientiousness were related to usage. In step 3, we entered Environmental personality. People who use public transport more frequently have a higher degree of Environmental personality compared to people who use public transportation less frequently.

### 5.4. The Effects of Personality Traits on Cycling

To examine the influence of personality traits on cycling we conducted still another hierarchical multiple regression analysis (see Table 5). Step 1 shows that high frequent cycling is related to being young, not having low income and presence of children at home. After controlling for these variables, the Big Five personality traits explained a significant amount of unique variance in cycling. Conscientiousness and Agreeableness are positively related to cycling. The effect from the Big Five personality traits disappears when Environmental personality is entered in step 3. The Environmental personality is explaining a significant amount of unique variance (*Adj. $R^2$* = 7%, $p < 0.001$). Worth noticing, Environmental personality explains more of cycling than the control variables and Big Five personality traits together (Table 5).

**Table 4.** Hierarchical regression models exploring how personality predict use of public transportation ($N$ = 959).

| Independent Variables | Step 1 Demography | Step 2 Demography, Big 5 [1] | Step 3 Demography, Big 5, EP [2] |
|---|---|---|---|
| Demography | | | |
| Gender (female) | 0.18 *** [3] | 0.17 *** | 0.15 *** |
| Age | −0.15 *** | −0.13 *** | −0.13 *** |
| Income (Low) | −0.02 | −0.03 | −0.03 |
| Income (High) | −0.04 | −0.04 | −0.04 |
| Child in household | −0.05 | −0.05 | −0.06 □ |
| Residential Gothenburg | 0.29 *** | 0.28 *** | 0.28 *** |
| Personality | | | |
| Openness | | 0.06 □ | 0.04 |
| Conscientiousness | | −0.06 □ | −0.07 * |
| Extraversion | | 0.02 | 0.02 |
| Agreeableness | | 0.09 ** | 0.08 ** |
| Neuroticism | | 0.03 | 0.02 |
| EP | | | 0.10 ** |
| $R^2_{Adj}$ | 0.156 *** | 0.168 *** | 0.175 *** |

[1] Big Five Personality traits, [2] Environmental personality. [3] Beta weights (i.e., standard regression coefficients), □ $p < 0.1$; * $p < 0.05$; ** $p < 0.01$; *** $p < 0.001$.

**Table 5.** Hierarchical regression models exploring how demography and personality predict cycling ($N$ = 954).

| Independent Variables | Step 1 Demography | Step 2 Demography, Big 5 [1] | Step 3 Demography, Big 5, EP [2] |
|---|---|---|---|
| Demography | | | |
| Gender (female) | −0.06 [3] | −0.08 * | −0.13 *** |
| Age | −0.11 *** | −0.10 ** | −0.09 ** |
| Income (Low) | −0.11 *** | −0.12 ** | −0.13 ** |
| Income (High) | 0.05 | 0.03 | 0.04 |
| Child in household | 0.08 * | 0.08 * | 0.06 |
| Residential Gothenburg | 0.02 | 0.01 | 0.02 |
| Personality | | | |
| Openness | | 0.03 | −0.02 |
| Conscientiousness | | 0.08 * | 0.05 |
| Extraversion | | 0.01 | 0.02 |
| Agreeableness | | 0.10 ** | 0.06 * |
| Neuroticism | | 0.05 | 0.03 |
| EP | | | 0.26 *** |
| $R^2_{Adj}$ | 0.049 *** (0.055) | 0.064 *** (0.075) | 0.122 *** (0.133) |

[1] Big Five Personality traits, [2] Environmental personality. [3] Beta weights (i.e., standard regression coefficients), * $p < 0.05$; ** $p < 0.01$; *** $p < 0.001$.

## 6. Discussion

### 6.1. Discussion on the Findings

The present study examined the effects of personality traits (i.e., the Big five personality traits and Environmental personality) on choice of transport modes (i.e., driving car, using public transportation, and cycling). First of all, we want to state that although personality traits make a significant contribution to explaining the use of different modes of transport, the effect of personality traits is quite modest, and comparable to previous research results [23]. The discussion here will focus on the significant effects ($p < 0.05$) of the third step of the regression models, which includes both the Big Five personality traits and Environmental personality.

### 6.1.1. Personality Traits and Car Driving

Car driving is positively related to Conscientiousness, and negatively related to Openness and Environmental personality. This indicates that more conservative people drive car often, since conservatism is primarily related to a low degree of Openness, but also to a high degree of Conscientiousness [12,13,17]. The car is probably perceived as a more time-efficient travel mode compared to public transport, and therefore preferred by people high on Conscientiousness. At least in Sweden, car drivers perceive the car as an efficient mode to carry out everyday activities [40].

Our finding regarding Openness is consistent with a representative study of the Swedish population [23]. However, there are also contradicting findings in some respects. The previous study found car driving positively related to Extraversion and negatively related to Neuroticism, while the present study did not find such relations. The difference between the two Swedish studies may be that the previous sample is population-representative in contrast to the sample in this study. This indicates some limitations in generalizing our findings outside Gothenburg, or perhaps outside urban areas. We suggest replications that investigate the relation between the Big Five personality traits and car driving, both inside and outside Sweden, in and outside urban areas

Our negative relation between Environmental personality and car driving is consistent with previous research, it shows that a low degree of Environmental personality increases the likelihood of choosing a mode with more carbon emissions [18].

### 6.1.2. Personality Traits and Use of Public Transportation

Use of public transportation is positively related to Agreeableness and Environmental personality, and negatively related to Conscientiousness. The relation between use of public transportation and Agreeableness correspond to findings representative for the Swedish population [23]. This might be explained by a friendly attitude toward other travelers and that public transportation is suitable for those who cooperate with others. The relation between public transportation and Environmental personality is consistent with previous research, which have found that a high degree of Environmental personality increases the likelihood of choosing an environmentally friendly mode [18]. High degree of Agreeableness and Environmental personality are positively associated with altruistic behaviors [15,17]. We suggest that future research investigate the relation between degree of altruism and use of public transportation. Regarding Conscientiousness, we argue in a similar way as for car driving. People with a high degree of Conscientiousness may perceive public transportation as less efficient and punctual, compared to their more easy-going and impulsive counterparts.

Our findings on the Big Five personality traits and usage of public transport are inconsistent with some previous research [1]. In contrast to a previous study, which found that use of public transport was positively related to Extraversion and negatively related to Neuroticism, we did not find any significant effects of these traits. The inconsistency might be explained by cultural differences between Sweden and Iran, and/or methodological differences in terms of scales and statistical analyses. A national representative study of the Swedish population, using the same personality scale as in the present study, found that public transportation is negatively related to Extraversion. Again, this indicates the limitations of generalizing the results of the present study.

### 6.1.3. Personality Traits and Cycling

Cycling is less explained by the Big Five personality traits, compared to car driving and usage of public transports. Cycling is better explained by the Environmental personality than the other two. The Environmental personality hence a higher association with frequency of cycling than any other factor in this study. In contrast to previous research, we found that women cycle as much as men do. First in steps two and three of the hierarchal regression analyses, when personality was entered, we found a gender effect, consistent with previous study–men cycle more than women [3]. It is worth noting that cycling is

related to a high degree of Conscientiousness and a high degree of Agreeableness. Further, it is interesting that both personality traits are explained by the Environmental personality (Table 5). Our interpretation of these analysis is that women in the Gothenburg area are cycling for environmental reasons more than men. We suggest more research on motives for men and women on cycling to understand what drives their behaviors.

*6.2. Limitiations of the Study*

The results of this study must be viewed considering its limitations. The cross-sectional design only permits exploration of relationships and not determining causal effects. From a theoretical point of view, it might be argued that personality traits cause the behavior and not the other way around [12,13].

Web-based surveys allow the involvement of many participants and may easily be used in Sweden where almost everyone has access to the Internet in their home or via smartphones [39]. However, one potential problem with our setting is that the sample is neither representative for Sweden nor the city of Gothenburg. Among our respondents, there was a higher proportion of men, older people, people with higher income and people without children in their households [34,37–39].

Another limitation is the self-reporting of choices of transport modes rather than measuring actual transport behavior. To estimate actual transports from introspective self-reports and questionnaire ratings cause problems familiar in the field of social and personality psychology [41]. We recommend future research to combine personality inventories with actual choices of transport modes, for instance through direct observations, automatic behavioral registrations, or geotagging through smartphones.

The Big Five personality traits were measured by a short scale to promote a higher response rate. This scale entails substantial losses and clear psychometric disadvantages in comparison to a full-length scale [35]. Further, the Environmental personality scale can be questioned in terms of construct validity. However, the internal consistencies for the five items of the Environmental personality is acceptable (Cronbach alpha = 0.7). It is also worth noting that the relations between the Big Five personality traits and the Environmental personality are consistent with previous research (Table 2, 19–21).

Although the Big Five personality traits show universal characteristics, modes of transport, and the relationship between personality traits and choice of modes of transport, might differ across regions [1,24]. Therefore, we cannot comment on the extent to which present and previous results might have been influenced by regional culture characteristics, related to Sweden in general and the urban area of Gothenburg in particular. In the light of our findings, future studies can replicate the study in different regions and cultures that may behave differently. The consistency with previous research regarding control variables strengthen the reliability of the present study. Frequencies in car driving is positively related to male, age, presence of child in the household, while frequencies in usage of public transportation is negatively related to male, age, and presence of child in the household [18,22,23,32].

## 7. Conclusions

The purpose of the study was to examine the influences of personality traits on the choice of transport mode between cars, public transport, and bicycles. More precisely, three research questions expressed the extent to which personality traits influence car driving, usage of public transportation, and cycling, respectively.

Personality traits are defined as the Big Five personality traits (i.e., Openness, Conscientiousness, Extraversion, Agreeableness and Neuroticism) and Environmental personality. Age, gender, income, children at home, and residential area are used as control variables in the analyzes. The study design was a survey, and the participants were a web-panel of adult citizen in the area of Gothenburg, Sweden (*N* = 1068).

Overall, the influence of personality traits on the frequency of use of different transport modes is small, especially regarding car driving and use of public transportation. Tradi-



tional and well-studied factors, such as age, income and residential area explains much more. Still, we want to draw some conclusions related to our purpose and research questions:

Some personality traits are the opposite between car drivers and people who use public transport. Car driving is influenced by a low degree of Openness, a high degree of Conscientiousness, and a low degree of Environmental personality. Use of public transportation is influenced by a high degree of Openness, a low degree of Conscientiousness, and a high degree of Environmental personality. Use of public transport is also influenced by a high degree of Agreeableness.

Cycling is foremost explained by a high degree of Environmental personality. Environmental personality explains cycling significantly better than the Big Five personality traits and the control variables used in this study. Moreover, environmental personality explains cycling better than it explains car driving and use of public transport.

The information presented here can be used by transportation planners and policy makers in understanding choices of different transport modes, especially if they want to change transport behaviors toward more sustainable options-from car driving to use of public transportation and cycling.

We believe that a better understanding of individuals can be used to change what lies outside these individuals–for example, infrastructure planning and design, information campaigns, and design of services. Better planning and design can facilitate behavioral changes. One identified challenge is to understand more about how behavioral change can occur among conservative people–those who drive car most frequently. Appealing to environmental effects might not be persuasive for car drivers, as one of our findings show. Another challenge is to design public transportation so that it appeals to people who have a high degree of Conscientiousness. This could for example, be carried out by improving reliability and ease of planning trips by public transport.

**Author Contributions:** Conceptualization and writing—original draft preparation, J.M.R., with contributions from F.S. and U.H.; methodology J.M.R. with contribution from F.S. and U.H. statistical analysis, J.M.R. review and editing, J.M.R., with contributions from F.S. and U.H. funding acquisition, F.S. and U.H. All authors have read and agreed to the published version of the manuscript.

**Funding:** The data collection was funded by The Swedish Research Council for Environment, Agricultural Sciences and Spatial Planning (Formas), grant number 2017-01029; the Swedish Energy Agency, grant number 44452-1.

**Institutional Review Board Statement:** All procedures were in accordance with the ethical standards of the institutional and/or national research committee and with the 1964 Helsinki declaration and its later amendments. Informed consent was obtained from all individual participants included in the study.

**Informed Consent Statement:** Informed consent was obtained from all subjects involved in the study.

**Data Availability Statement:** The data presented in this study are available on request from the corresponding author.

**Acknowledgments:** We would like to thank the Laboratory of Opinion Research (LORE); University of Gothenburg, for their help with data collection. We would like to thank Tommy Gärling and Cecilia Jakobsson Bergstad for useful comments.

**Conflicts of Interest:** The authors declare no conflict of interest.

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
