# Peer review of "Traits and Transports: The Effects of Personality on the Choice of Urban Transport Modes"

_applsci, doi:10.3390/app12031467_

Round 1
Reviewer 1 Report
The paper deals with the theme of the influence of personality in the choice of the type of means of transport between cars, public transport, and bicycles. The work is well written, and the methodology and steps are well explained. The paper analyses in detail even the most similar references to the work done, comparing the results. Added values are to have given ideas for future research and underlined the limits of the work.
Here below some minor comments to improve the quality and effectiveness of the paper:
Minor comment 1
How this article contributes to the existing literature is well explained in section 3. I suggest mentioning it also in the introduction (section 1) to give a complete interpretation of the work.
Minor comment 2
Section 1, line 55, insert change in "research question" in line with the previous two.
Minor comment 3
Section 2, line 57, there is a typo in the word personality. To be corrected.
Minor comment 4
Section 6, lines 286 and 342, when one i.e. it is good to insert a comma like (i.e., ....)
Minor comment 5
The conclusions turn out to be a bit poor compared to the work done. I suggest enriching this section by inserting a summary of all the previous paragraphs. The introduction and conclusions should be able to be read on their own and explain the work carried out in all its parts.
Minor comment 6
For all text, follow the paper template, with the same text size (section 3), and remove the yellow highlight.
Author Response
Please, see the Word file.

Reviewer 2 Report
General Remark // Thank you for sharing this interesting piece of work, which I genuinly enjoyed reading. I personally think, with some additional work on restructuring and clarification in the introduction, methodology and discussion section this can be an intersting addition to the field of research. Good look with this! :)
Title
I would suggest to specify the Title in accordance with the abstract and use “personality traits” instead of personality, as ‘just’ referring to personality sounds a little bit like a generalization in a tabloid rather than scientific manner
Introduction + Definition + Previous research
General remarks //
- Content // In the beginning of the introduction, I am missing a little bit of an upbuild in the Introduction to catch the reader. E.g regarding the relevance of the studying mode of transportation behaviour, followed by previous research on transportation behaviour that focusses on travel time and cost, then followed by the idea to look into personality due to car usage research, …. ,
- Content // At the End of the introduction if would suggest to also shortly explain, why (on what grounds) you choose the other two modes besides car usage (most common ? most interesting ? most relevant ? randomly ? ), just to give the reader an idea about the development of all three questions.
- Structure // I general, I would consider revising the chapter organization of the first three chapters. This is just an open proposal, but I have the impression that chapter 2 and 3 can also be seen as part of the derivation of the research questions: ‘Previous research on personality and mode of transportation focusses on the environmental personality…This can be defined as … There is only little known about the relationship between other personality theories, like the big five.. the big five can be defined as… Consequently, we developed the research question xyz.. Additionally we included other variables that can influence personality and mode of transport, as research shows that… xxx‘
- Content // After reading the introduction, I was not sure in how far the environmental personality and the BIG 5 are related to one another and whether you intend to investigate both or just the BIG 5 . Some of the confusion might stem from the statement of your focus in line 61 on 62 . In any case I would suggest to point it the relationship between those theories very clearly to the reader, as well as your intention to study both.
Specific remarks //
- 53 // I think a “research” is missing here
- 59 // I would suggest to add some more sources here, as this is a very wide and general statement ( “variety of domains”, “stable set of characteristics”) (maybe it is also possible to refer to some of the sources that are used in the upcoming paragraphs on the big five – but that is just an assumption, I have not checked them ? )
Materials and Methods
General remarks //
- 147 and ff // Structure // Personally, I find it a bit confusing to summarize sample and procedure in one subheading. I would rather use two headings for this , as I was not sure after my first read, what your procedure actually looked like.
- Also, I am also missing some information about the design of your study
- Lastly, I am missing information about the statistical analyses you intend to do to answer your question
Specific remarks //
162 – 163 // content // I was a little confused by this sentence, as you write in the beginning of the paragraph that the panel itself is not representative in relation to age and gender. How did you achieve this?
189 -191 // Content // what do you mean by “you were influenced”? This might be a translation thing, but I was not quite sure what was meant by this.
216 – 217 // Content // another reference to the representativeness. I would suggest to focus on describing the attributes of your sample (mean of age, mean of male/female participants) rather than focussing on the representativeness. Especially since the panel is not representative and your sample was not as well. And maybe add one sentence at the end somewhere that this is not representative for the Gothenburg population
Discussion
General remarks //
- Structure // I got a little bit lost half way through the discussion. I would suggest to semantically structure the discussion a little bit more clearly: starting with the most relevant results, In accordance with the research questions, followed by the most relevant interpretation of these result in the light of the context of other results or alternative explanations.
- Content // While reading the discussion, I realized again that the distinction between EP and Big 5 and your hypothesis in relation to this is not perfectly clear to me.
Specific remarks //
- 345 // Title // I think, I would call this subheading ‘limitations of the study’ rather than ‘methodological considerations’.
- 345 – 347 // Content // Is it scientifically sound though to assume that personality traits cause the behaviour based on a cross-sectional-design? Wouldn’t it also be possible that alternative variables that are not considered in the design could explain both ? I would suggest to abstain from this assumption as the luring implication is not really covered by the methodological design.
- 380- 382 // Content // I would propose to include this in the general discussion rather than the methodological considerations
Author Response
Please, see the attached file.

Reviewer 3 Report
Interesting study. Sound data--within the limitations indicated by the authors. The authors mention potential impacts on transport policy. However, this aspect of the study needs more elaboration. How can we use this information to promote sustainable transportation choices?
Also, a number of grammatical and spelling errors throughout.
Author Response
Please, see the attached file.

Round 2
Reviewer 2 Report
Dear authors,
thank you very much for the neat and well prepared cover letter. This made it very easy for me to review your paper again.
I do not have any additional commentary and wish you all the bes for your publication.
This manuscript is a resubmission of an earlier submission. The following is a list of the peer review reports and author responses from that submission.
Round 1
Reviewer 1 Report
The present article investigates the influence of demography and personality on urban travel mode preferences. The article has many serious major flaws, therefore I strongly suggest the rejection of the article. My comments are as below:
- I doubt that the authors have good knowledge in the collection, organization, analysis, interpretation, and presentation of data. Methodological reasoning is missing in this article as well as the used applied methods are not suitable for this article, furthermore none of the used methods are explained. The authors probably conducted Pearson correlation analysis even though they do call it just as ‘’correlation’’. There are four types of correlation in literature: Pearson correlation, Kendall rank correlation, Spearman correlation, and the Point-Biserial correlation. How do you think that Pearson correlation analysis is suitable for this study when you have dichotomous and ordinal variables?
- The presented correlation results do not have any scientific value, the authors tell that x correlated with y, y correlated with z, etc. There is almost no correlation between the majority investigated variables, what kind of correlation you talking about? The presented correlation tables show that almost all of the Pearson correlation coefficients are below 0.3.
- The authors also presented modeling attempts with the uncorrelated variables. The presented models do not mean anything scientifically, R2 and beta coefficients are too small to consider. As well as, there are not any model fitting information and multicollinearity analyses presented for any model.
- There is also a big disbalance in the distribution of some demographical variables. Such as gender - %61 of the survey attendees are men, around % 50 of the survey attendees have in the middle-income group. The distribution disbalance can cause strange results in the statistical analysis.
- Looking into literature review related sections of the article, many important works in the field of travel behavior were neglected. The literature review does not contain information such as which method was used in the cited study, when and where this study was conducted, etc. The effect of transport geography on urban travel habits is excluded.
- Looking from the point of the article topic, the authors tell from the title that they investigate ‘’demography’’. From the point of the demographical investigation, there are more than hundreds of works based on the influence of demography on urban travel mode preferences. Why we need one more? If we will have one more, this one should provide a very deep analysis and should demonstrate very particular results. But this study does not provide this. When we mention demography, this doesn't mean only age, gender, income, and presence of children. There are many other demographical factors excluded in this study such as level of education, employment status, profession, marital status, household composition. I would not consider demography for this study. ''What could be interested to study'' is the assessment of personality variables considering Swedish cultural identity on urban travel mode preferences (an example related to the effect of cultural identity on urban travel mode preferences: https://doi.org/10.3390/socsci8080227)
- There are not any conclusions presented for this study, I do not see that there is any gain of this study to the current literature.
Reviewer 2 Report
This paper deals with the effects of demography and personality on urban travel habits: the frequency of usage of cars, bicycles and public transportation. The Authors use a web-based survey among the non-representative sample collected from the Citizen Panel at the University of Gothenburg. The hierarchical multiple regression analysis is addressed to examine the influence of demography and personality of urban travel decisions. The work has the advantage of some pro-environmental overtones.
The topics covered in the article are interesting and promising, but I have fundamental comments about the content of the article:
- The abstract seems to be a bit too detailed and miswritten
- The article only considers a non-representative example on a local scale. Why has there been no attempt to achieve representativeness, at least on a city-wide basis? How should these results be read in the context of wider society?
- The city of Gothenburg is wealthy and, by world standards, fairly clean, with a socially sensitive population. This is good, of course. However, it seems that these results would be different in a broader population. There was a lack of discussion in this direction, beyond the general wording that 'it could be different.
- The literature study on psychology-related issues is broader, than the one on transportation issues. The latter requires substantive development. This is important because we don't know how much personality factors prevail, for example, on economic or geographical issues. This is a part of a bigger problem: the transportation issues have generally been treated in a minor way, compared to psychological issues.
- The authors themselves acknowledge that ``The magnitudes of the relationship between the Big Five personality factors and habits reported in this study is quite modest'. This is a rather elegant and diplomatic statement, but one with rather obvious implications.
Besides, the article appears to be slightly disorganized. It contains recurring themes, and some issues are left without further explanation. Moreover, some language issues need to be reviewed (preferably by a native speaker).
In summary, I consider the article to have numerous methodological flaws that, in this version, preclude publication in this journal.
Reviewer 3 Report
The literature review should be expanded, especially in the area of ​​the influence of character on travel habits. Questions asked to respondents should be described in detail. It would be worth trying to present the results on the charts for a more confident interpretation.